# Immune Response Gaps Linked to SARS-CoV-2 Infection: Cellular Exhaustion, Senescence, or Both?

**DOI:** 10.3390/ijms232213734

**Published:** 2022-11-08

**Authors:** Leonardo Vinicius Barbosa, Daniele Margarita Marani Prá, Seigo Nagashima, Marcos Roberto Curcio Pereira, Rebecca Benicio Stocco, Francys de Luca Fernandes da Silva, Milena Rueda Cruz, Djessyka Dallagassa, Thiago João Stupak, George Willian Xavier da Rosa Götz, Georgia Garofani Nasimoto, Luiz Augusto Fanhani Cracco, Isabela Busto Silva, Karen Fernandes de Moura, Marina de Castro Deus, Ana Paula Camargo Martins, Beatriz Akemi Kondo Van Spitzenbergen, Andréa Novais Moreno Amaral, Caroline Busatta Vaz de Paula, Cleber Machado-Souza, Lucia de Noronha

**Affiliations:** 1Postgraduate in Biotechnology Applied in Health of Children and Adolescent, Faculdades Pequeno Príncipe (FPP), Instituto de Pesquisa Pelé Pequeno Príncipe (IPPPP), R. Silva Jardim, 1632 Água Verde, Curitiba 80250-060, Brazil; 2Postgraduate Program of Health Sciences, School of Medicine, Pontifícia Universidade Católica do Paraná (PUCPR), R. Imaculada Conceição, 1155 Prado Velho, Curitiba 80215-901, Brazil; 3Laboratory of Experimental Pathology, School of Medicine, Pontifícia Universidade Católica do Paraná (PUCPR), R. Imaculada Conceição, 1155 Prado Velho, Curitiba 80215-901, Brazil; 4Hospital Marcelino Champagnat, Av. Presidente Affonso Camargo, 1399 Cristo Rei, Curitiba 80050-370, Brazil

**Keywords:** SARS-CoV-2, inflammation, pathogenesis, cellular exhaustion, senescence, CD8, perforin

## Abstract

The COVID-19 pandemic, promoted by the SARS-CoV-2 respiratory virus, has resulted in widespread global morbidity and mortality. The immune response against this pathogen has shown a thin line between protective effects and pathological reactions resulting from the massive release of cytokines and poor viral clearance. The latter is possibly caused by exhaustion, senescence, or both of TCD8+ cells and reduced activity of natural killer (NK) cells. The imbalance between innate and adaptive responses during the early stages of infection caused by SARS-CoV-2 contributes to the ineffective control of viral spread. The present study evaluated the tissue immunoexpression of the tissue biomarkers (Arginase-1, CCR4, CD3, CD4, CD8, CD20, CD57, CD68, CD138, IL-4, INF-α, INF-γ, iNOS, PD-1, Perforin and Sphingosine-1) to understand the cellular immune response triggered in patients who died of COVID-19. We evaluated twenty-four paraffin-embedded lung tissue samples from patients who died of COVID-19 (COVID-19 group) and compared them with ten lung tissue samples from patients who died of H1N1pdm09 (H1N1 group) with the immunohistochemical markers mentioned above. In addition, polymorphisms in the Perforin gene were genotyped through Real-Time PCR. Significantly increased tissue immunoexpression of Arginase, CD4, CD68, CD138, Perforin, Sphingosine-1, and IL-4 markers were observed in the COVID-19 group. A significantly lower immunoexpression of CD8 and CD57 was also found in this group. It is suggested that patients who died from COVID-19 had a poor cellular response concerning viral clearance and adaptive response going through tissue repair.

## 1. Introduction

The rapid spread of coronavirus disease (COVID-19) has resulted in widespread global morbidity and mortality, causing an unprecedented public health crisis worldwide [1]. As in the Influenza A virus H1N1 subtype pandemic (H1N1pdm09), the clinical manifestations of COVID-19 are heterogeneous. Patients range from asymptomatic or mild upper respiratory disease to severe viral pneumonia, evolving to acute respiratory distress syndrome (ARDS) and death [2,3].

Growing evidence indicates that a hyper-inflammatory response to SARS-CoV-2 virus contributes to critical cases and decease outcomes in COVID-19 (1). Like other respiratory viral diseases, the adaptive response phase, especially T-cell responses, plays a relevant role in this new coronavirus infection [4].

The defense against pathogens is mediated by innate and late adaptive immunity [5]. Each consists of different cell types and distinct functions [6]. In a generic example of viral infection, the immune system quickly recognizes the entry of an extrinsic agent through cells and effector molecules of the innate immune system [5]. Macrophages, monocytes, dendritic cells and neutrophils express several pattern recognition receptors (PRRs), identifying the invasive pathogen’s characteristics [7]. Such external molecular structures are known as pathogen-associated molecular patterns (PAMPs). The binding of PAMPs to PRRs triggers the inflammatory response against the viruses by expressing transcription factors that induce pro-inflammatory cytokines, such as the nuclear kappa B factor (NF-kB). This binding also induces activation of crucial components of immediate antiviral response, such as type I Interferons (alpha—α and beta—β), which are crucial to limit viral replication and dissemination [5,8].

Type I Interferons increase the cytotoxicity of natural killer cells (NK), which kill infected cells through a mechanism similar to those of CD8+ T cytotoxic lymphocytes (CD8+ CTLs) [5]. These cells carry granules containing proteins, such as perforin and granzymes, that support the death of target cells. In addition, NK cells produce interferon-gamma (INF-γ) that recruits macrophages to destroy phagocytosed microorganisms [9]. Type I interferons also promote differentiation from immature T cells to Th1 helper T cell subgroups. It also upregulates histocompatibility molecules, increasing the probability of infected cells being recognized by CD8+ CTLs [10].

The cytokines produced by antigen-presenting cells (APC) against viral infections stimulate the activation, proliferation and differentiation of lymphocytes during the innate response, besides influencing the nature of the adaptive response [4]. Once activated, naive CD4+ T differs in effector cells that can promote B-cell activation. These B-cells can assist CD8+ T cells to play an antiviral role due to the production of IFN-γ, among other cytokines, and activate phenotype type I macrophages [6]. Macrophage activation is the primary function of the immune response mediated by T cells to eliminate intracellular microorganisms through phagocytosis and induce a robust inflammatory response [11].

In addition to the Th1 subtype, CD4+ T helper lymphocytes can differentiate into the Th2 subtype. This differentiation is mediated mainly by Interleukin 4 (IL-4), which also induces the activation of M2 phenotype macrophages, which is crucial to the tissue repair process [12]. The polarization process of CD4+ T cells is mediated via transcription factors, chemokines and cytokines, where regulation of inflammatory molecules associated with cells can lead to the onset and increase of inflammatory lung diseases such as ARDS [13].

In persistence or chronic exposure to antigens, such as chronic viral infections, CD8+ T cell responses can be gradually suppressed by the expression and coupling of inhibitive receptors such as Programmed Cell Death Protein 1 (PD-1) [5]. Exhausted CD8+ T cells express PD-1, which blocks CLTS activation and induces decreased production of cytokines such as INF-γ [14]. In addition, advanced age contributes to an issue of immunological senescence, implying reduced plasticity of CD4+ T and CD8+ T and a compromised adaptive response against viral infections [15].

Specifically, in events of SARS-CoV-2 infection, its immunopathology is still subjective. Although the scientific community has elucidated several issues regarding COVID-19, questions about cellular immunity still need clarification. In this sense, the present study analyzed twenty-four post-mortem biopsies of patients affected by COVID-19, aiming to comprise the behavior of the cellular response and genomic polymorphisms, especially CD8+ T cells and their molecular secretion content, concerning their exhaustion, senescence, or both.

## 2. Results

The relevant demographic, clinical, histopathological analyses and immunohistochemistry findings of the groups are summarized in Table 1. Data of comorbidities, laboratory findings at admission, laboratory tests 24 h before death, and drugs administered during hospitalization are described in Appendix A.

The gender-related demographic analysis did not reveal statistically significant results. However, there is a considerable involvement of diseases concerning male patients in both groups. Concerning age-related data, there is a statistically significant result for the COVID-19 group compared to the H1N1 (*p* < 0.001), considering a predominance of deaths of elderly patients in the former. It is also noticeable that the data referring to the time from hospitalization to death and the use of mechanical ventilation demonstrated significantly increased results in the COVID-19 group (*p* = 0.003 and *p* = 0.028, respectively). Regarding pre-existing lung diseases, four of the twenty-four patients in the COVID-19 group presented bronchial asthma and one presented interstitial pulmonary fibrosis. The histopathological findings show interstitial pneumonitis with slight neutrophilic exudation in the COVID-19 group. In addition, the hyaline membrane, type II pneumocyte hyperplasia, fibrosis, and micro thrombosis were present. Regarding the H1N1 group, interstitial pneumonitis with exudation of septal neutrophils and no microthrombi are observed.

COVID-19 group demonstrated significantly increased tissue immunoexpression of Arginase-1 and IL-4 (*p* = 0.002 and *p* < 0.0001, respectively) and increased number of Perforin-1+ (*p* = 0.009), CD4+ (*p* = 0.009), CD68+ (*p* = 0.013), CD138+ (*p* < 0.0001) cells in comparison to the H1N1 group. A significantly decreased number of CD8+ and CD57+ (*p* = 0.03 and *p* = 0.001, respectively) cells were observed in the COVID-19 group compared to the H1N1 group. No statistical differences were observed in the tissue immunoexpression of CCR4 (*p* = 0.845), INF-α (*p* = 0.637), INF-γ (*p* = 0.806), iNOS (*p* = 0.610) and Sphingosine-1 (*p* = 0.06), nor in number of CD3+, CD20+ (*p* = 0.533 and *p* = 0.416, respectively) (Figure 1 and Figure 2) and PD-1+ (*p* = 0.07) cells in the COVID-19 group when confronted to the H1N1 group.

The distribution of alleles and MAF frequencies for all tag SNPs for the *PRF1* gene can be observed in Appendix A.

## 3. Discussion

### 3.1. Demographic and Histopathological Findings

The demographic findings referring to the COVID-19 and H1N1 groups corroborate with those found in the literature [16], considering that the COVID-19 group was primarily composed of elderly patients (average of 71.96). It is worth mentioning that the samples that comprise this study were obtained in March and August 2020, when deaths from COVID-19 mainly affected older adults. These patients also carried relevant comorbidities and remained hospitalized for long periods and undergoing mechanical ventilation. On the other hand, the age of patients in the H1N1 group (average of 43.5), hospitalization periods, and mechanical ventilation exposure were lower, as described in the literature [17]—Table 1.

Diffuse alveolar damage (DAD) with intra-alveolar hyaline membrane was observed in both groups in evaluating histopathological aspects. Studies that described the histopathological patterns of lung injury in patients affected by COVID-19 and H1N1pdm09 report the existence of DAD [18]. Moreover, patients in the COVID-19 group also presented terminal fibrosis, microthrombi and type II pneumocyte hyperplasia. Pulmonary fibrosis is generally present in COVID-19 fatal outcomes and is the pathognomonic aspect of the repair process in ARDS [19].

Shreds of evidence suggest the involvement of pyroptosis in the endothelial dysfunction induced by SARS-CoV-2 infection contributing to the micro thrombosis observed in the COVID-19 group, in contrast to H1N1pdm09 [20]. Differently from the COVID-19 group, a septal neutrophilic infiltrate is recruited in the H1N1 group cases. Influenza infection increases the host’s susceptibility to bacteria, mainly due to a failure of the antibacterial immune response and the synergy of pathogens [21].

### 3.2. Cellular Response

SARS-CoV-2, like other viruses that infect humans, has immune system evasion mechanisms [6]. Studies have demonstrated that this virus effectively delays the triggering process of innate immune system responses [6,22].

Due to the host infection, cells associated with the innate immune system recognize the viral molecular, the PAMPs. Phagocytes, such as macrophages and epithelial cells, express PRRs and bind to PAMPs, activating signal transduction pathways that promote anti-inflammatory functions, among other functions [5].

Hence, evaluating the immunoexpression of CD68 results, it is observed that it was significantly increased in the COVID-19 group (*p* = 0.013). Depletion of resident alveolar macrophages and a concomitant increase in monocyte-derived inflammatory macrophages have been previously demonstrated in patients stricken by COVID-19 [23].

Mononuclear phagocytes stimulate the expression of type I Interferons, with INF-α being the most important in viral defense [5]. When infected by viral agents, other cell lineages also secrete INF-α to signal adjacent cells, producing viral proteins that recognize and degrade exogenous genetic material [24]. No statistical difference was observed in tissue immunoexpression INF-α between both groups in this study. This result is plausible since it is a comparison between two viruses.

It is known that type I interferons increase the cytotoxicity of NK cells and CD8+ CTLs and promote the differentiation of immature T cells to TH1 helper T cell subgroups. The present study observed a decrease in NK cell number (CD57) in the COVID-19 group. Witkowski et al. [25] evidenced critically COVID-19 patients with high serum levels of transforming growth factor β (TGF-β), which reached peaks during the first two weeks of infection. This premature production of TGF-β could inhibit NK cell function and viral control. In addition, the present study’s average survival time of patients in the H1N1 group was 4.7 days. In comparison, the average survival time of the COVID-19 group was 15.87 days, which means that patients in the H1N1 group presented in the acute phase of the disease by the time of death.

When evaluating CD3 immunoexpression, no significant difference was observed between the COVID-19 and H1N1 groups. However, evaluating the CD4 immunoexpression noted a significantly increased difference in the COVID-19 group. There is evidence that CD4+ CTLs infiltrate/expand in the lung parenchyma of critically ill COVID-19 patients, and this infiltration/expansion is more prominent in the disease resolution phase. The increase of inflammation accompanies these augmentations in infiltration/expansion—however, late CD4+ CTLs can positively contribute to the issue of viral cleansing. However, in some contexts of the disease, it can contribute negatively, possibly by providing an immunological trigger for the development of fibrosis [26]. This data also corroborates with the study since, in the histopathological criteria, terminal fibrosis was identified in COVID-19 patients.

In viruses, CD4+ T cells usually differentiate into the Th1 subtype, which has antiviral activity through the production of INF-γ, thus, capable of M1 phenotype macrophage recruitment [6]. These macrophages, when activated by INF-γ, produce reactive nitrogen species, primarily nitric oxide (NO), through the action of the induced nitric oxide synthase (iNOS) enzyme, which holds a relevant role in the destruction of microorganisms [27]. Regarding tissue immunoexpression of INF-γ and iNOS, no statistically significant difference was observed when comparing the groups. However, evaluating the immunoexpression of CCR4, IL-4, Sphingosine-1 and Arginase, it is deduced that the COVID-19 group had the Th2 pathway activated. This group had significantly increased values of IL-4 and Arginase-1 (Table 1), in addition to the Sphingosine-1 increased value trend. Additionally, this finding corroborates with the literature since the functions of Th2 cells are mediated principally by IL-4, whose association is compatible with worse prognoses [28]. Cytokines that stimulate the Th2 pathway to interfere with the Th1 response by suppressing the activation of macrophages with M1 phenotype stimulate the production of TGF-β, which is inefficient in viral clearance [16].

Effector CD4+ T cells respond to antigens by producing cytokines with functions of recruitment and activation of leukocytes and B lymphocytes. In addition to being aimed at producing specific neutralizing antibodies, B lymphocytes play an essential role in APCs once they express class II MHC [27]. Albeit the immunoquantitation of B lymphocytes (CD20) did not present a statistically significant result, its differentiated form, plasma cells (CD138), showed a statistically increased result in the COVID-19 group compared to H1N1 (Table 1). It should be taken into account that, although the quantification of differentiated B lymphocytes is high, the virus, after infecting a cell, becomes inaccessible to the antibody, and its eradication will require CD8+ T lymphocytes (CTLs) to kill the infected cells and eliminate the reservoir of infection [5].

The cellular response to SARS-CoV-2 is overwhelming, especially regarding CD8+ T cells. CD8+ CTLs recognize peptides derived from protein antigens associated with class I MHC and eventually kill virus-infected cells by producing perforins and granzymes molecules [29]. The tissue immunoexpression of CD8+ is statistically reduced in the COVID-19 group, although perforin is statistically increased in this group. In addition, it is noticed a statistical trend of PD-1 in the COVID-19 group (Table 1). Some main points should be reported and discussed. First, as previously mentioned, the average age of patients in the COVID-19 group is 71.96, which would explain the low cellular immunoquantitation of CD8+. Advanced age is a common condition regarding the severity of respiratory viral diseases, which could be associated with altered T-cell responses. Cellular senescence may also be associated with age, contributing to ineffective responses to viral infection [4]. The T cells may reach a replicative senescence stage due to the telomerase activity disablement [30].

Associated with senescence, cellular exhaustion must also be considered. Although no statistical significance of PD-1 was observed, this marker presented a statistical trend for the COVID-19 group. Whereas functional effector T cells transiently express inhibitory receptors, persistent pathogen stimulation can induce reduction and exhaustion of T cell function during chronic infection [31]. Exhausted CD8+ T cells show functional and phenotypic changes, including increased expression of inhibitory receptors such as PD-1 [5,32]. Recently, critically ill COVID-19 patients were reported to have a state that transitioned from hyperactivation to exhaustion of CD8+ T cells, with high expressions of PD-1 [4]. In addition, TGF-β may also be associated with T cell exhaustion. TGF-β promotes T cell dysfunction or exhaustion during chronic viral infection [33]. The findings of the present work predict that the Th2 pathway is activated in the COVID-19 group; due to this activation, there is a secretion of TGF-β, intending to promote tissue remodeling.

In addition, a recently published study by our research group showed an increased presence, but not statistically significant, of TGF-β in patients affected by COVID-19 compared to those affected by H1N1pdm09, reinforcing the participation of TGF-β pathways in the development process of pulmonary fibrosis. It should be taken into account that patients who died from H1N1pdm09 also present histopathological findings compatible with DAD [16]. Fujita et al. suggested that the presence of TGF-β resulted from the successive regeneration and proliferation of alveolar cells caused by influenza A/H1N1pdm09 viral infection [34].

However, the association of cell exhaustion and elevated TGF-β expression in COVID-19 needs further elucidation.

If, on the one hand, CD8+ T cells are senescent, exhausted, or both, on the other hand, in this study, they seem to be hyperactivated. The quantification of perforin in the COVID-19 group was significantly increased compared to the H1N1 group. Perforin (and granzyme)-induced apoptosis is the main pathway for cytotoxic lymphocytes (including NK cells) to eradicate virus-infected host cells [35]. Perforin is a protein that forms pores in the membrane of target cells, allowing the entry of effector molecules and subsequent cell death [36].

### 3.3. Perforin Polymorphism

In a recent article by our group, Zanchettin et al. suggested that perforin, although present, may be dysfunctional. This aspect may also be associated with polymorphisms in the perforin gene [37] in the COVID-19 group. Polymorphisms in the perforin gene can produce non-functional proteins, which could cause more expression in an attempt to compensate for the ineffectiveness of inducing apoptosis. The *PRF1* rs885822 G/A presents a 56.8% for the wild G allele in the COVID-19 group, and this frequency is almost twice that observed in two databases (Appendix A). The G wild allele was associated with overall survival in childhood acute lymphoblastic leukemia. Patients carrying the GG genotype demonstrated a significantly higher risk of death than those carrying the A allele [38]. This observation may be a mechanism associated with the process of immune T cell exhaustion.

Scientific evidence suggests that SARS-CoV-2 can trigger a poor immune response, especially in patients with advanced age. The present study emphasizes that SARS-CoV-2 infection can compromise the cellular response and poor viral clearance due to a state of senescence, cellular exhaustion, or both. The adaptive response is tended toward tissue repair. An association of immunohistochemistry and polymorphisms could be an asset in identifying potentially susceptible individuals, which may evolve to worsen outcomes.

Furthermore, it must be considered that the challenges for new treatment modalities for COVID-19 affect several systems ranging from the chronic inflammatory process to lung tissue repair caused by SARS-CoV-2.

### 3.4. Study Limitations

This study is a retrospective analysis; samples were obtained from post-mortem biopsies. Thus, the information from this cohort cannot reconstruct the events in a chronological evolution of the disease regarding previous data concerning clinical or physiopathological aspects but only provide a momentary perspective of the outcome. Considering the small sample, the results inferred in this study are preliminary, suggesting that future studies with more extensive and age-paired samples be carried out.

## 4. Materials and Methods

### 4.1. Ethical Approval

The presented study was approved by the National Research Ethics Committee (Conselho Nacional de Ética em Pesquisa—CONEP) under protocol numbers 3.944.734/2020 and 2.550.445/2018. The authors confirm that all methods were carried out following relevant guidelines and regulations. The families permitted the post-mortem biopsy of the cases of COVID-19 and H1N1pdm09 and signed the informed consent forms.

### 4.2. Samples

COVID-19 group (*n* = 24): Post-mortem lung samples from patients who died of SARS-CoV-2 came from the Intensive Care Unit (ICU) of the Marcelino Champagnat Hospital in Curitiba, Brazil, comprising infection from March to August 2020. A minimally invasive lung biopsy was performed through a left anterior mini-thoracotomy with upper left lobe segment resection. The resected pieces measured 3 × 3 cm. The time of lung sample acquisition after death was less than 2 h. Clinical data were obtained from medical records during hospitalization in the Intensive Care Unit (ICU) at the Marcelino Champagnat in Curitiba, Brazil. Testing for COVID-19 was performed by nasopharyngeal swabs taken during ICU hospitalization and performed Real-time Polymerase Chain Reaction (RT-qPCR). The viral genome amplification was performed with the Invitrogen SuperScript™III Platinum^®^ One-Step qRT-PCR Kit (Catalog number: 11732020, Waltham, MA, USA), which was positive for SARS-CoV-2.

H1N1 group (*n* = 10): Lung samples from patients who died at the Intensive Care Unit (ICU) of Hospital de Clínicas de Curitiba, Brazil, of H1N1pdm09 were obtained by minimally invasive lung post-mortem biopsy (COVID-19 similar technique). The patients were tested for H1N1pdm09 through the qRT-PCR (COVID-19 similar technique).

### 4.3. Immunohistochemistry Analysis

The immunohistochemistry assay was preceded by preparing multisample paraffin tissue blocks, TMA (Tissue Microarray). The representative areas of the lung were previously demarcated and identified. Then, three cylindrical fragments measuring 0.3 cm in diameter were extracted from the original formalin-fixed paraffin-embedded blocks (FFPE donor blocks) and compiled into new TMA blocks.

The immunohistochemistry technique was applied to identify the immunoexpression of Arginase-1, CCR4, CD3, CD4, CD8, CD20, CD57, CD68, CD138, IL-4, INF-α, INF-γ, iNOS, PD-1, Perforin-1 and Sphingosine-1 as shown in Table 1 and Appendix A.

The technique recommended an overnight incubation protocol for primary antibodies. The secondary polymer (Mouse/Rabbit PolyDetector DAB HRP Brown, BSB0205, BioSB, Santa Barbara, CA, USA) was applied to the tested material for 40 min at room temperature. The revelation took place with the addition of the complex 2, 3, diamino-benzidine + hydrogen peroxide substrate for a brown color turning time and later, the counterstaining with Harris Hematoxylin was performed. The reactivity of positive control confirmed the results. A known immunoexpression tissue sample to be positive for the antibody in question was allocated together with the samples studied.

The slides of Arginase-1, CCR4, IL-4, INF-α, INF-γ and iNOS were scanned with Axio Scan.Z1 Scanner (ZEISS, Jena, Germany). Then, the software ZEN Blue Edition (ZEISS, Jena, Germany) was utilized for the generation of 30 high power fields (HPF, 40× magnification) (COVID-19 group) and 20 HPF (H1N1 group) randomly. The analysis was blind once the software randomly generated the images, with no observer interference. The immunopositivity areas were measured by the software Image-Pro Plus 4.5 (Media Cybernetics, Rockville, MD, USA). Subsequently, these areas were converted into percentages.

Slides immunostained with CD3, CD4, CD8, CD20, CD57, CD68, CD138, PD-1 and Perforin-1 were used for: lymphocyte count, NK cell count, PD-1 receptor count and pore-forming protein in CD8+ T and NK cells, respectively. Only immunoexpressed cells were counted at 20 HPF, in the alveolar septa and perivascular spaces, with the optical microscope BX 50 (OLYMPUS, Tokyo, Japan). Afterward, the arithmetic mean values of the 20 HPF of each patient were calculated, and their results were organized and submitted for statistical analysis.

The slides immunostained by Sphingosine-1 were observed under an optical microscope and analyzed in 10 HPF, using the scoring method known as Allred Score. This method evaluates the proportion and intensity of immunopositivity of M2 macrophages and type II pneumocytes. The semiquantitative analysis was obtained by summing two scores (proportion and intensity of positivity), ranging from 0 to 8. The proportion score was subdivided according to the percentage of cellular immunoexpression, in which cell score could be score 0–0% stained cells, score 1: < 1%, score 2: 1–10%, score 3: 11–33%, score 4: 34–66% and score 5: >66%. While the intensity of positivity was evaluated: negative: score 0, weak: score 1, moderate: score 2, and strong: score 3.

### 4.4. Genetic Alleles Analysis

The authors used the data obtained in a recent paper [39] of our group that involved perforin analysis in obtaining the alleles data for the *PRF1* gene. The allelic frequency was compared with the NCBI site’s two databases (TOPMED and 1000G) [39].

### 4.5. Statistical Analysis

The normality condition was evaluated using the Shapiro–Wilk test. The nonparametric for the continuous immunohistochemical variables between two groups was performed using the Mann–Whitney test and was characterized by the median, interquartile range, and minimum and maximum values. The parametric for the continuous demographic and clinical variables between two groups, performed using the Student’s *t*-test, were characterized by mean and standard deviation values. For categoric variables, the performed test was Fisher’s exact test, and its values were characterized by absolute number and frequency. Values of *p* < 0.05 indicated statistical significance. The data were analyzed using the software JMP™ Pro 14.0.0. (SAS Institute, Cary, NC, USA). The genotype was expressed by absolute number and frequency, and the logistic regression model test was performed to acquire the *p*-values. The data were analyzed using the computer program by IBM^®^ SPSS Statistics v.20.0 software (IBM, Armonk, NY, USA). 

## Figures and Tables

**Figure 1 ijms-23-13734-f001:**
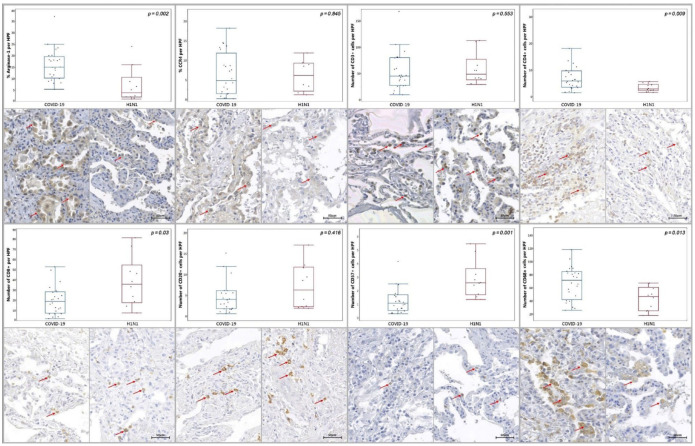
Graphs demonstrating tissue immunoexpression of Arginase-1 (percentage per HPF) and the number of immunopositive cells per HPF of CCR4, CD3+, CD4+, CD8+, CD20+, CD57+ and CD68+ in the alveolar epithelium in the COVID-19 and H1N1 groups. Red arrows denote the immunoexpression of each tested marker.

**Figure 2 ijms-23-13734-f002:**
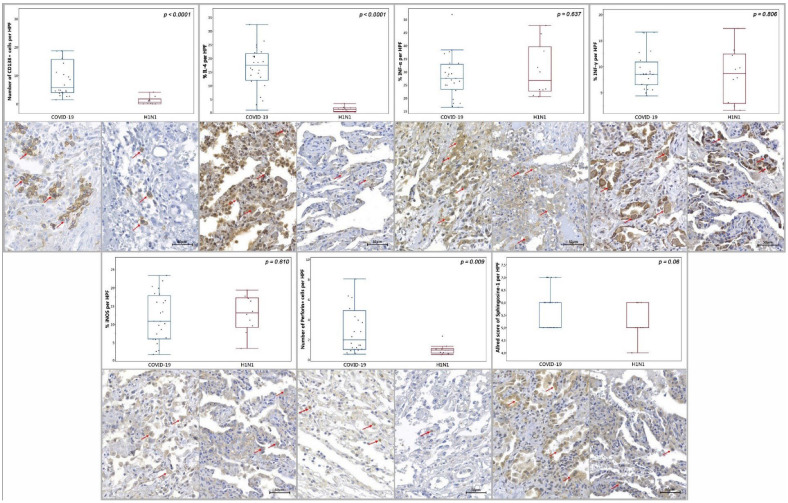
Graphs demonstrating tissue immunoexpression of IL-4, INF-α, INF-γ, iNOS (percentage per HPF), the number of immunopositive cells per HPF of CD138+ and Perforin+ and the Allred score of Sphingosine-1 in the alveolar epithelium in the COVID-19 and H1N1 groups. Red arrows denote the immunoexpression of each tested marker.

**Table 1 ijms-23-13734-t001:** Comparison between COVID-19 and H1N1 groups according to demographic, clinical, histopathological and immunohistochemistry findings.

Data	COVID-19 (*n* = 24)	H1N1 (*n* = 10)	*p*-Value
**Gender ^a^**	Male 15 (62.5%) Female 9 (37.5%)	Male 8 (80%) Female 2 (20%)	0.437 *
**Age (years) ^b^**	71.96 ± 12.5	43.5 ± 14	<0.001 **
**Time from hospitalization** **to death (days) ^b^**	15.87 ± 10.2	4.70 ± 6.13	0.003 **
**Mechanical ventilation ^b^**	12 ± 9.2	4.7 ± 6.13	0.028 **
**Previous pulmonary** **diseases**	Bronchial Asthma (4/24) Interstitial Pulmonary Fibrosis (1/24)	-----	-----
**Histopathological findings**	Interstitial pneumonitis with scarce septal neutrophils, hyaline membrane, type II pneumocyte hyperplasia, fibrosis, and micro thrombosis	Interstitial pneumonitis with high septal neutrophils infiltration and no micro thrombosis	-----
**Tissue Arginase-1 ^c^**	14.98/9.36 (5.32–37)	3.81/ 8.60 (1.1–24)	0.002 ***
**Tissue CCR4 ^c^**	4.77/10.34 (0.19–18.18)	6.14/8.06 (1.19–11.86)	0.845 ***
**Number of CD3+ ^d^**	45/52.8 (9.3–168)	48.55/38.92 (29–112.45)	0.533 ***
**Number of CD4+ ^d^**	5.9/6.14 (1.45–56.4)	2.8/2.32 (1.55–5.65)	0.009 ***
**Number of CD8+ ^d^**	18.8/20.77 (1.9–52.65)	35.7/82.84 (7.5–81.05)	0.03 ***
**Number of CD20+ ^d^**	4.02/4.4 (0.6–24.95)	6.3/9.51 (1.85–17.05)	0.416 ***
**Number of CD57+ ^d^**	1.08/1.15 (0.3–4.3)	2.58/1.92 (1.35–5.45)	0.001 ***
**Number of CD68+ ^d^**	71.52/42.53 (25.75–118.05)	46.7/35.54 (17.8–67.15)	0.013 ***
**Number of CD138+ ^d^**	5.78/12.75 (1.45–57.85)	0.58/1.75 (0–4.1)	<0.0001 ***
**Tissue IL-4 ^c^**	17.5/9.72 (1.02–32.4)	1.36/1.41 (0.36–3.41)	<0.0001 ***
**Tissue INF-α ^c^**	10.14/14.5 (0.52–32.75)	10.35/7.54 (1.36–20)	0.637 ***
**Tissue INF-γ ^c^**	8.5/4.39 (4.36–16.62)	8.66/9.5 (1.57–17.34)	0.806 ***
**Tissue iNOS ^c^**	10.88/11.81 (1.78–23.35)	13.14/8.08 (3.43–19.36)	0.610 ***
**Number of PD-1+ ^a,d^**	24 (72.7 %)	9 (27.3 %)	0.07 *
**Number of Perforin-1+ ^d^**	1.98/3.89 (0.55–12)	0.95/0.44 (0.5–2.35)	0.009 ***
**Allred of Sphingosine-1 ^e^**	6/1 (5–7)	5/1 (4–6)	0.06 ***

**Legend:** Continuous variables are expressed by median/interquartile range (minimum–maximum). ^a^ Categorical variables expressed by absolute number and (frequency). ^b^ Demographic variable age (years) and clinical variables time from hospitalization to death (days) and time of mechanical ventilation (days) expressed by mean ± standard deviation. ^c^ Tissue expression in percentage per HPF. ^d^ Number of cells per 20 HPF (average). ^e^ Allred Score. * Fisher’s exact test. ** Student’s *t*-test. *** Mann–Whitney test.

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
