# Peer review of "Immune Response Gaps Linked to SARS-CoV-2 Infection: Cellular Exhaustion, Senescence, or Both?"

_ijms, 2022, doi:10.3390/ijms232213734_

Round 1

Reviewer 1 Report

This manuscript describes the histopathological analyses and immunohistochemistry results of 24 patients died from COVID-19, with 10 patients died from H1N1 infection as control. Though the patients died early in 2020, which made this study not a timely report, the histopathological and immunohistochemistry data from these patients would provide supportive evidence to indicate that immunopathological mechanisms of COVID-19 differs from that of H1N1 infection. A concern of this study is that the COVID-19 patients were significantly much older than H1N1 patients and aging may weaken the comparative conclusions. Besides, all the discussion were based on histopathological and immunohistochemistry analyses, and comprehensive conclusions may need to adopt multiple testing results. Overall, the results from this manuscript are valuable for our understanding of the immunopathological mechanisms of COVID-19, especially when compared with the infection of H1N1.

There are some suggestions for the authors:

1.     Clinical data of these patients was not well presented in this manuscript. A timeline of infection and disease course of each of the patients infection with SARS-CoV-2 or H1N1 virus should be described or presented as a figure. The disease course and treatment of the patients should be described in detail in Results section.

2.     Laboratory results for the subjects should be included in Table 1, if available, including the viral load, blood biochemical examination and whole blood test results, etc. The longitudinal variation of the clinical and laboratory values should be presented as a figure, if it is accessible.

Author Response

  1. Clinical data of these patients was not well presented in this manuscript. A timeline of infection and disease course of each of the patients infection with SARS-CoV-2 or H1N1 virus should be described or presented as a figure. The disease course and treatment of the patients should be described in detail in Results section.

Thank you for this consideration. We created a supplementary table 2 with the following data from the COVID-19 group:

Comorbidities,

Computed tomography chest at admission,

Relevant initial laboratory tests,

Laboratory tests 24 hours before death and,

Therapeutic drugs.

We do not have such data for patients in the H1N1 group.

Considering that the present work discusses inflammatory cell findings, we chose to demonstrate them as a supplementary table 2.

We emphasize that all patients in both groups tested positive for qRT-PCR.

  1. Laboratory results for the subjects should be included in Table 1, if available, including the viral load, blood biochemical examination and whole blood test results, etc. The longitudinal variation of the clinical and laboratory values should be presented as a figure, if it is accessible.

Thank you for this consideration. Laboratory results: Relevant initial laboratory tests, Laboratory tests 24 hours before death are described in supplementary table 2.

We do not have such data for patients in the H1N1 group.

Reviewer 2 Report

Dear Author,

The work is very interesting but overall the topic is not explored in the best possible way.

The main point in favor of the work is undoubtedly the casuistry as having post-mortem lung biopsy samples from Covid-19 patients is a considerable resource; to date, the immune responses in the lungs of COVID-19 patients remain poorly characterized. 

The main disadvantage is the discussion is unclear and sometimes controversial and needs to be better developed.

Major revision:

1# I suggest increasing the biopsy data by adding control patients. In your previous study published in Viruses (https://doi.org/10.3390/v14081699) you performed analysis on the same samples, but in this study, you also added a series of control biopsies which however were not mentioned in this document.

2# Therefore I propose to repeat the same analysis on the control samples, but also to add the analysis with the TGF-β marker which becomes of great importance in this work, as also pointed out by you on line 248.

3# In your previous work, immunohistochemical analysis of perforin, CD8 +, and CD57 + is performed and in the discussion, it was stated that "the histopathological analysis showed high tissue expression of perforin in the COVID-19 group; CD8 + was high in the H1N1 group and CD57 + in the CONTROL group ". In light of the previous results and of the latter, for which you suggest that deceased Covid-19 patients have an inefficient immune response, or at least a TH2-type immune phenotype (line 222), how do you justify the faster death of patients with infection with H1N1 viruses showing a greater CD8 + T response, with a mature/ TEMRA CD57 + phenotype, a lower expression of PD-1 molecule associated with cell exhaustion and lower perforin values?

4# In the discussion (lines 259-263) the previous study is mentioned but it is not declared that it is the same group that conducts the experiments.  I find the sentence below deliberately ambiguous, as it seems that the group that performed the experiments is a different group: “Zanchettin et al. analyzed the perforin tissue expression and the SNPs of the PRF1 gene in patients stricken by a severe form of COVID-19 and patients affected by H1N1. They observed, through a model of correlation between the protein expression and its genetic polymorphisms, that, like our results, the highest tissue expression values of perforin were associated with the genotyping in PRF1 in the COVID-19 diseased patients [38] ". Moreover, in the discussion, you repeat what was already published in the previous paper and I find this inappropriate especially because it is not the focus proposed in the title. The supplementary data proposed in the supplementary material tab. 2 are the same already published in the paper overcited. 

It is clear to me that you want to give strength to what has been done, but I suggest declaring from the beginning that this is an extension of the previous study with the aim of confirming, expanding, and better investigating the impact of the local immune response on the control of Covid-19 infection.

4# Finally, I suggest including in your paper, these two article references. The first is a  study published in Nature communication, in which two different patterns of immune response to Covid-19 are identified: one pattern shows a high local expression of interferon and cytokines, high viral loads, and limited pulmonary damage, the other pattern shows severely damaged lungs, low Interferon, low viral loads and abundant infiltrating activated CD8 + T cells and macrophages. The first group died earlier after hospitalization than the second group of patients. https://doi.org/10.1038/s41467-020-18854-2.  The second one is the following paper: DOI work: https://doi.org/10.1016/j.chest.2020.09.259 "Lung Histopathology in Coronavirus Disease 2019 as Compared With Severe Acute Respiratory Syndrome and H1N1 Influenza A Systematic Review". I believe these papers could be very useful for your discussion.

Minor revision

-line 181: DAD, the acronym has never been mentioned before, so Diffuse Alveolar Damage must be added

-Figures 1 and 2 .: The Y axis reference is missing in the box plots at the top left (probably  due to a layout problem)

Author Response

The main disadvantage is the discussion is unclear and sometimes controversial and needs to be better developed.

Major revision:

1# I suggest increasing the biopsy data by adding control patients. In your previous study published in Viruses (https://doi.org/10.3390/v14081699) you performed analysis on the same samples, but in this study, you also added a series of control biopsies which however were not mentioned in this document.

We appreciate and understand your suggestion.

The CONTROL group mentioned by the esteemed reviewer is composed of lung samples from patients who do not present any histopathological alteration in the lung tissue since they died of cardiovascular and neoplastic diseases. Lung samples from these patients do not show any pathological cellular infiltrates in the intra-alveolar spaces or in the interstitium of the alveolar septa. The inflammatory cells present are only the rare basal alveolar lymphocytes and macrophages. We chose not to add them, considering that the present study's focus is to understand the cellular response by analyzing the cellular infiltrates and samples from patients whose lung tissue did not have any pathological alterations that would not have such an impact on the cellular response in the pulmonary tissue.

On the other hand, comparing these data with lung samples affected by a respiratory disease of viral etiology and causing a pandemic leads us to understand the questions still not clarified about the cellular infiltrates involved in the viral immune response.

We created a supplementary table 2 with the following data from the COVID-19 group:

Comorbidities,

Computed tomography chest at admission,

Relevant initial laboratory tests,

Laboratory tests 24 hours before death and,

Therapeutic drugs.

We do not have such data for patients in the H1N1 group.

Considering that the present work discusses inflammatory cell findings, we chose to demonstrate them as a supplementary table 2.

2# Therefore I propose to repeat the same analysis on the control samples, but also to add the analysis with the TGF-β marker which becomes of great importance in this work, as also pointed out by you on line 248.

We understand your point; however, as mentioned in the previous answer, the comparison with H1N1 samples allows us to elucidate the issues involving inflammatory cell infiltration in COVID-19. We omitted TGF- β results in this article because it was recently published in an article by our group. Thus, we chose to include only in our discussion a study carried out by our group, whose comparison of TGF-β was shown to be increased in patients affected by COVID-19, emphasizing that this cytokine seems to be involved in the tissue remodeling process, as well as the activated TH2 pathway.

As described in the paper, we emphasize that the association of TGF-β and CD8 needs to be better understood.

3# In your previous work, immunohistochemical analysis of perforin, CD8 +, and CD57 + is performed and in the discussion, it was stated that "the histopathological analysis showed high tissue expression of perforin in the COVID-19 group; CD8 + was high in the H1N1 group and CD57 + in the CONTROL group ". In light of the previous results and of the latter, for which you suggest that deceased Covid-19 patients have an inefficient immune response, or at least a TH2-type immune phenotype (line 222), how do you justify the faster death of patients with infection with H1N1 viruses showing a greater CD8 + T response, with a mature/ TEMRA CD57 + phenotype, a lower expression of PD-1 molecule associated with cell exhaustion and lower perforin values?

Some comments to answer this question:

  • 1- Although viruses cause respiratory diseases, their etiology, pathophysiology, epidemiology, and etiopathogenesis differ. The immune response may or may not be related to the death process as the latter is the result of several factors such as comorbidities, pattern and severity of diffuse alveolar damage, acute course of the disease, chronicity, presence of systemic inflammatory disease, individual constitutional factors, viral clearance capacity, effective therapeutic measures, among others. Several of these factors could be influenced directly or indirectly by the TH1-type response, others not so much. Thus, the early death of patients with H1N1 seems more related to a faster and more acute course of the disease with a more rapidly evolving diffuse alveolar damage with severe edema. This fact may be related to a more robust inflammatory response since the patients were younger than COVID-19 patients.
  • The expected immune response to a viral pathogen should occur, above all, via CD8+. In other words, the CD8+ lymphocyte count in the cases of the H1N1 group was expected. It turns out that, as with COVID-19, patients affected by H1N1pdm09 could also develop a cytokine storm and an exacerbated Th17 response with a vast pulmonary neutrophilic inflammatory infiltrate.
  • The average survival time of patients in the H1N1 group is 4.7 days, while that of the COVID-19 group is 15.9 days. That is, it is suggested that because of a delay in the innate response, in addition to the viral replication being more intense, thus tissue damage is also more significant and more prolonged.
  • The average age of patients in the COVID-19 group is 71.96 years, that is, elderly patients whose cellular immune response may be senescent or in the process of exhaustion.

4# In the discussion (lines 259-263) the previous study is mentioned but it is not declared that it is the same group that conducts the experiments. I find the sentence below deliberately ambiguous, as it seems that the group that performed the experiments is a different group: "Zanchettin et al. analyzed the perforin tissue expression and the SNPs of the PRF1 gene in patients stricken by a severe form of COVID-19 and patients affected by H1N1. They observed, through a model of correlation between the protein expression and its genetic polymorphisms, that, like our results, the highest tissue expression values of perforin were associated with the genotyping in PRF1 in the COVID-19 diseased patients [38] ". Moreover, in the discussion, you repeat what was already published in the previous paper and I find this inappropriate especially because it is not the focus proposed in the title. The supplementary data proposed in the supplementary material tab. 2 are the same already published in the paper overcited. 

It is clear to me that you want to give strength to what has been done, but I suggest declaring from the beginning that this is an extension of the previous study with the aim of confirming, expanding, and better investigating the impact of the local immune response on the control of Covid-19 infection.

The author changed the genetic approach as we thought it necessary to reinforce the findings of the recent article by our group. We used allele frequency analysis. The authors also sought to compare the frequencies found with MAF (Minimum Allelic Frequency) records from two databases present for rs885822 on the NCBI website.

4# Finally, I suggest including in your paper, these two article references. The first is a  study published in Nature communication, in which two different patterns of immune response to Covid-19 are identified: one pattern shows a high local expression of interferon and cytokines, high viral loads, and limited pulmonary damage, the other pattern shows severely damaged lungs, low Interferon, low viral loads and abundant infiltrating activated CD8 + T cells and macrophages. The first group died earlier after hospitalization than the second group of patients. https://doi.org/10.1038/s41467-020-18854-2. The second one is the following paper: DOI work: https://doi.org/10.1016/j.chest.2020.09.259 "Lung Histopathology in Coronavirus Disease 2019 as Compared With Severe Acute Respiratory Syndrome and H1N1 Influenza A Systematic Review". I believe these papers could be very useful for your discussion.

Thanks for your suggestions. The study published in Nature is already included in our work.

Minor revision

-line 181: DAD, the acronym has never been mentioned before, so Diffuse Alveolar Damage must be added

Added.

-Figures 1 and 2 .: The Y-axis reference is missing in the box plots at the top left (probably  due to a layout problem)

Solved.

***Changes are highlighted in yellow.

Round 2

Reviewer 1 Report

all concerns addressed

Reviewer 2 Report

The authors have clarified my doubts and added what's suggested. The manuscript is now more straightforward and complete.